# Epidemiology of prediabetes mellitus among hill tribe adults in Thailand

**Tawatchai Apidechkul**[1]*, **Chalitar Chomchiei**[1], **Panupong Upala**[1],
**Ratipark Tamornpark**[1,2]

**1** Center of Excellence for The Hill Tribe Health Research, Mae Fah Luang University, Chiang Rai, Thailand,
**2** School of Health Science, Mae Fah Luang University, Chiang Rai, Thailand

* Tawatchai.api@mfu.ac.th

**Data Availability Statement:** All relevant data are within the paper and its Supporting Information files.

**Funding:** Tawatchai Apidechkul was granted for doing this project by The Health System Research

## Abstract

### Background

Prediabetes is a major silent health problem that leads to the development of diabetes within a few years, particularly among those who have a low socioeconomic status. Hill tribe people are vulnerable to prediabetes due to their unique cultural cooking methods and their hard work on farms, as well as their low economic status and educational levels. This study aimed to estimate the prevalence of prediabetes among hill tribe people in Thailand and identify the related factors.

### Methods

This cross-sectional study included participants who belong to one of the six main hill tribes: Akah, Lahu, Hmong, Yao, Karen, and Lisu. The study was conducted in 30 hill tribe villages in Chiang Rai Province, Thailand. A validated questionnaire was administered, and 5-mL blood specimens were collected. Data were collected between November 2019 and March 2020. Logistic regression was used to determine the associations between independent variables and prediabetes.

### Results

A total of 1,406 participants were recruited for the study; 67.8% were women, 77.2% were between 40 and 59 years old, and 82.9% were married. The majority worked in the agricultural sector (57.2%), had an annual income ≤ 50,000 baht (67.5%), and had never attended school (69.3%). The prevalence of prediabetes was 11.2%. After controlling for age and sex, five factors were found to be associated with prediabetes. Members of the Akha and Lisu tribes had 2.03 (95% CI = 1.03–3.99) and 2.20 (95% CI = 1.10–4.42) times higher odds of having prediabetes than Karen tribe members, respectively. Those with hypertension (HT) had 1.47 (95% CI = 1.03–2.08) times higher odds of having prediabetes than those with normal blood pressure. Those with a normal total cholesterol level had 2.43 (95% CI = 1.65–3.58) times higher odds of having prediabetes than those with a high total cholesterol level. Those with a high triglyceride level had 1.64 (95% CI = 1.16–2.32) times higher odds of having prediabetes than those with a normal triglyceride level. Those with a high low-

Institute, Thailand (Grant No 61-027). However, the funder had no role in study design, data collection and analysis, decision to publish, or preparation of the manuscript.

**Competing interests:** The authors have declared that no competing interests exist.

density lipoprotein cholesterol (LDL-C) level had 1.96 (95% CI = 1.30–2.96) times higher odds of having prediabetes than those with a normal LDL-C level.

## Conclusion

Appropriate dietary guidelines and exercise should be promoted among hill tribe people between 30 and 59 years old to reduce the probability of developing prediabetes.

## Introduction

Prediabetes is a silent health problem worldwide [1]. With only minor signs and symptoms, a large proportion of people with prediabetes are not diagnosed or cared for properly [1]. The Centers for Disease Control and Prevention (CDC) in the United States established the criterion for prediabetes as a hemoglobin A1c (HbA1c) level of 5.7–6.4% [2]. Yip et al. [3] reported the prevalence of prediabetes, with different rates in different populations, for example, 13.5% in Caucasian populations and 18.2% in Asian populations, by using combined impaired fasting glucose (IFG) and impaired glucose tolerance (IGT) methods. Notably, 5.0%–10.0% of people with prediabetes will develop diabetes mellitus (DM) each year [4]. This imposes a tremendous burden on the health care system, particularly due to high annual medical expenses [5], and reduces the quality of life of individuals with overt diabetes [6]. This problem will more than double among people with a poor economic status, especially in developing countries.

The Ministry of Public Health, Thailand, reported that the overall prevalence of prediabetes among Thai individuals 15 years and older was 10.7% (11.8% among men and 9.5% among women) [7]. The prevalence was high among people 30 years and older. Those who had a low socioeconomic status had low rates of proper diagnosis and care [1, 2]. The Thai government allocates a large amount of money yearly to care for patients with noncommunicable diseases, including diabetes [8]. Identifying people with prediabetes in a community, especially among those with a poor socioeconomic status, would be greatly advantageous for the design and implementation of proper interventions to reduce the rate of DM development or delay its onset.

Hill tribes comprise populations of people who have migrated from southern China to the northern region of Thailand over the past few centuries [9, 10]. There are six main groups: Akha, Lahu, Hmong, Yao, Karen, and Lisu [10]. Almost all have their own culture, language, and lifestyle, which are close to those of Chinese people, except the Karen, who originated from the Thailand-Myanmar border [10]. Today, more than 70.0% of these individuals have been granted Thai identification cards, which indicate Thai citizenship, are used to access all public services, including medical care [11, 12]. However, there is no scientific information available about prediabetes among the hill tribe people in Thailand. Thus, the objectives of this study were to examine the prevalence of prediabetes among hill tribe populations over 30 year old living in northern Thailand and identify the related factors.

## Methods

### Study design/study setting

A community-based cross-sectional study was performed to gather information from participants who lived in 30 hill tribe villages located in 18 districts in Chiang Rai Province, Thailand. In 2019, there were 749 hill tribe villages in Chiang Rai Province, which included 316 Lahu

villages (51,339 persons), 243 Akha villages (74,403 persons), 63 Yao villages (16,227 persons), 56 Hmong villages (33,478 persons), 36 Karen villages (7,933 persons), and 35 Lisu villages (9,632 persons) [12].

## Study population and eligible population

The study population comprised hill tribe people who belonged to one of the six main tribes: Akah, Lahu, Hmong, Yao, Karen, and Lisu. Those who lived in the 30 selected hill tribe villages (five villages from each tribe) and were between 30 and 59 years old met the inclusion criteria. However, those who were previously diagnosed with DM, who could not provide essential information according to the study protocol, and who did not comply with the no food or beverage (NPO) instructions for the 12 hours before blood sample collection were excluded from this study.

## Sample size calculation

The sample size was calculated according to the standard formula for a cross-sectional study as a proportion [13], n = $[Z^2\alpha/_2 {}^*P^*Q]/e^2$, where Z = the value of the standard normal distribution corresponding to the desired confidence level (Z = 1.96 for 95% CI), P = the expected true proportion, which was based on a previous study conducted among Thai adults, at 32.2% [14], and e = the desired precision, or percentage of the accepted deviation, which was 6.00%. Therefore, at least 1,398 participants were required for analysis.

After the sample size was calculated, five villages from each tribe were randomly selected by a computer-generated randomization method as shown in the following flowchart (Fig 1). Afterward, all people aged 30–59 years living in the selected villages were invited to participate in the study. All people were screened according to the inclusion and exclusion criteria before the initiation of data collection.

## Research instrument and its development

The questionnaire was developed and tested before use. The questionnaire was divided into three parts. In part one, twelve questions were used to collect sociodemographic information, such as age, sex, education, tribe, religion, and occupation. In part two, three questions were used to collect information about exercise, alcohol consumption and smoking. In the last part, eight open-ended questions were used to collect information about the physical examination and laboratory results, such as weight, height, blood pressure, and lipid profiles.

Item-objective congruence (IOC) was applied to improve the validity of the questionnaire. Using this method, three external experts (one medical doctor, one epidemiologist, and one public health professional) were invited to comment on the relevance of the questions in the questionnaire and the context of the study, including the objective of the study. There were three options provided to score each question. The questions were scored as -1 if they were not related to the content of the study, 0 if they required revision before use, and +1 if they reflected the content of the study. Afterward, the scores for each question assigned by the three experts were summed and divided by three. The number obtained was used to decide whether to retain the question in the questionnaire set. If the summed score was less than 0.5, the question was removed from the questionnaire. Questions scoring 0.5–0.7 were revised according to the comments before being included in the questionnaire. Questions scoring more than 0.7 were considered acceptable to include in the questionnaire.

Afterward, the questionnaire was piloted among 20 hill tribe people who had characteristics similar to those of the study population. In this step, we aimed to detect its reliability, the proper sequence of questions, and the ability of the target population to understand the

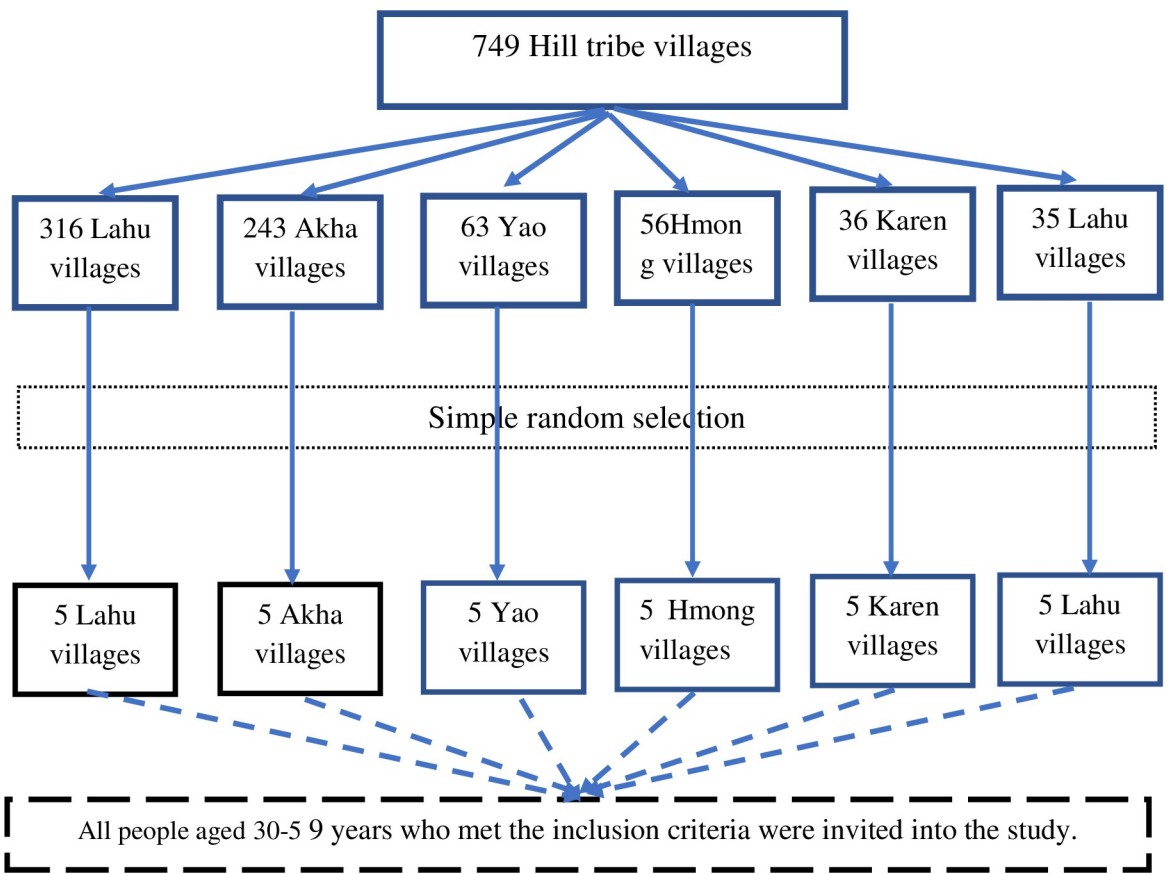

**Fig 1. Flowchart of sample selection in the study.**

questions. The pilot study was performed in a hill tribe village in Mae Chan District, Chiang Rai Province.

## Operational definitions

Prediabetes was defined as an HbA1c level between 5.7% and 6.4% according to the CDC guidelines [2]. Body mass index (BMI) was classified into three categories: equal to or less than 18.50 was defined as underweight, 18.51–22.99 was defined as normal weight, and equal to or higher than 23.00 was defined as overweight [15]. Hypertension (HT) was classified as systolic blood pressure (SBP) equal to or greater than 140 mmHg, diastolic blood pressure (DBP) equal to or greater than 90 mmHg, or both [16]. Total cholesterol was classified into two categories: normal (<200 mg/dL) and high (≥200 mg/dL) [17]. High-density lipoprotein cholesterol (HDL-C) was classified into two categories for males, normal (≥40 mg/dL) and low (<40 mg/dL), and two categories for females, normal (≥50 mg/dL) and low (<50 mg/dL) [17]. LDL-C was classified into two categories: normal (<100 mg/dL) and high (≥100 mg/dL) [17]. Triglycerides were classified into two categories: normal (<150 mg/dL) and high (≥150 mg/dL) [17].

## Data collection procedures

Five villages from each tribe were randomly selected from a list of the relevant hill tribe villages in Chiang Rai Province by a computer-generated method. Access to the villages was granted

by district government officers. All selected village headmen were contacted 5 days prior to the date of data collection to provide essential information regarding the study, especially the target population and the inclusion and exclusion criteria. One day before the research team reached the village, the village headman informed all participants about the 12-hour NPO prior to blood specimen collection. On the date of data collection, each of the participants was provided all essential information again before providing voluntary informed consent. Those who were able to read and write in Thai completed the forms by themselves. However, those who could not understand Thai were helped by village health volunteers who were fluent in both Thai and the local language. Completion of the questionnaire and the collection of 5-mL blood specimens took approximately 25 minutes each. Data were collected between November 2019 and July 2020.

## Laboratory work

All laboratory work was performed at the Mae Fah Luang Medical Laboratory Center. The latex-enhanced immunoturbidimetric method (RANDOX$^{\circledR}$) was used to measure HbA1c; this method has been certified by the National Glycohemoglobin Standardization Program and standardized to the Diabetes Control and Complication Trial reference. All lipid profiles, i.e., total cholesterol, HDL-C, and LDL-C levels, were assessed by the direct clearance method. Tri-glyceride levels were measured by the glycerin phosphate oxidase peroxidase method.

## Statistical analysis

All completed questionnaires were coded and entered into an Excel sheet. The data were checked and managed for errors and missing data before being uploaded to the SPSS program (version 24, Chicago, IL) for analysis. Means and standard deviations are presented for continuous data with a normal distribution, while medians and interquartile ranges (IQRs) are presented for continuous data with a skewed distribution. Logistic regression was used to detect the association at a significance level of $\alpha = 0.05$ in both the univariate and multivariate models. The "ENTER" mode was used to select independent variables in the model, and the final model was shown by the Hosmer Lemeshow Chi-square test to be suitable. Before interpreting the final model in the multivariate analysis, age and sex were controlled for as confounding factors.

## Ethics approval

All of the study concepts and the protocol were approved by the Mae Fah Luang University Research Ethics Committee on Human Research (No. REH-6100) before project commencement. Participants were provided all essential information before providing written informed consent. For those who could not understand Thai, village health volunteers provided the information in the local language before asking these participants to voluntarily provide a fingerprint representing informed consent.

# Results

## General characteristics of the participants

A total of 1,406 participants were recruited for the study; 67.8% were female, 77.2% were between 40 and 59 years old (mean = 46.1, SD = 8.0), and 82.9% were married. The majority had never attended school (69.3%), worked in the agricultural sector (57.2%), and had an annual income ≤ 50,000 baht (67.5%), with a median of 30,000 baht (IQR = 44,500). Some participants reported that their parents had been diagnosed with DM: 5.3% of fathers and 7.3% of mothers (Table 1).

One-fourth (24.8%) of the participants smoked, 26.5% used alcohol, 60.0% did not exercise, 19.9% had moderate-to-high stress, and 10.1% reported depressive symptoms. A large proportion were classified as overweight (64.2%), 45.2% had high total cholesterol, 40.7% had high triglycerides, and 69.8% had high LDL-C (Table 1).

**Prevalence of prediabetes.** One hundred fifty-eight (11.2%) out of 1,406 participants had an HbA1c level between 5.7% and 6.4%, indicating prediabetes (Table 2). The prevalence did not differ according to sex (p-value = 0.357), age (p-value = 0.273), or tribe (p-value = 0.066).

## Factors associated with prediabetes

In the univariate analysis, five factors were found to be associated with prediabetes: tribe, exercise, total cholesterol, LDL-C, and HT (Table 2).

**Table 1. General characteristics of the participants.**

| Factors | n | % |
|---|---|---|
| **Total** | **1,406** | **100.0** |
| **Sex** | | |
| Male | 453 | 32.2 |
| Female | 953 | 67.8 |
| **Age** (years) | | |
| 30–39 | 321 | 22.8 |
| 40–49 | 538 | 38.3 |
| 50–59 | 547 | 38.9 |
| **Marital status** | | |
| Single | 86 | 6.1 |
| Married | 1,166 | 1,166 |
| Ever married | 154 | 11.0 |
| **Family members** (people) | | |
| ≤ 4 | 722 | 51.4 |
| 5–8 | 586 | 41.6 |
| ≥9 | 98 | 7.0 |
| **Tribe** | | |
| Karen | 225 | 16.0 |
| Akha | 408 | 29.0 |
| Lahu | 236 | 16.8 |
| Hmong | 198 | 14.1 |
| Yao | 191 | 13.6 |
| Lisu | 148 | 10.5 |
| **Religion** | | |
| Buddhist | 716 | 50.9 |
| Christian or Muslim | 690 | 49.1 |
| **Education** | | |
| Never attended a school | 975 | 69.3 |
| Primary school | 250 | 17.8 |
| Secondary school and higher | 181 | 12.9 |
| **Occupation** | | |
| Unemployed | 190 | 13.5 |
| Agriculturist | 776 | 55.2 |
| Daily employment or trader | 440 | 31.3 |

(*Continued*)

**Table 1.** (Continued)

| Factors | n | % |
|---|---|---|
| **Total** | **1,406** | **100.0** |
| **Annual income** (baht) | | |
| ≤ 50,000 | 949 | 67.5 |
| 50,001–100,000 | 338 | 24.0 |
| ≥ 100,001 | 119 | 8.5 |
| **Family debt** | | |
| No | 862 | 61.3 |
| Yes | 544 | 38.7 |
| **Paternal DM history** | | |
| No | 894 | 63.6 |
| Yes | 75 | 5.3 |
| Do not know | 437 | 31.1 |
| **Maternal DM history** | | |
| No | 895 | 63.6 |
| Yes | 102 | 7.3 |
| Do not know | 409 | 29.1 |
| **Smoking** | | |
| No | 1,057 | 72.5 |
| Yes | 349 | 24.8 |
| **Alcohol consumption** | | |
| No | 1034 | 73.5 |
| Yes | 372 | 26.5 |
| **Exercise** | | |
| No | 844 | 60.0 |
| Sometimes | 460 | 32.7 |
| Regularly | 102 | 7.3 |
| **BMI** | | |
| Normal weight | 434 | 30.9 |
| Underweight | 69 | 4.9 |
| Overweight | 903 | 64.2 |
| **Hypertension** | | |
| No | 995 | 70.8 |
| Yes | 411 | 29.2 |
| **Total cholesterol** | | |
| Normal | 771 | 54.8 |
| High | 635 | 45.2 |
| **Triglycerides** | | |
| Normal | 834 | 59.3 |
| High | 572 | 40.7 |
| **HDL-C** | | |
| Normal | 636 | 45.2 |
| Low | 770 | 54.8 |
| **LDL-C** | | |
| Normal | 424 | 30.2 |
| High | 982 | 69.8 |

**Table 2. Factors associated with prediabetes mellitus in the univariate and multivariate logistic regression analyses.**

| Factors | Prediabetes mellitus | | Univariate analysis | | | Multivariate analysis | | |
|---|---|---|---|---|---|---|---|---|
| | Yes n (%) | No n (%) | OR | 95% CI | p value | AOR | 95% CI | p value |
| Total | 158 (11.2) | 1,248 (88.8) | N/A | N/A | N/A | N/A | N/A | N/A |
| **Sex** | | | | | | | | |
| Male | 56 (35.4) | 397 (31.8) | 1.18 | 0.83–1.67 | 0.358 | 0.90 | 0.55–1.46 | 0.677 |
| Female | 102 (64.6) | 851 (68.2) | 1.00 | | | 1.00 | | |
| **Age** (years) | | | | | | | | |
| 30–39 | 39 (24.7) | 282 (22.6) | 1.00 | | | 1.00 | | |
| 40–49 | 67 (42.4) | 471 (37.7) | 1.03 | 0.68–1.57 | 0.896 | 0.99 | 0.62–1.60 | 0.989 |
| 50–59 | 52 (32.9) | 495 (39.7) | 0.76 | 0.49–1.18 | 0.221 | 0.79 | 0.47–1.30 | 0.355 |
| **Marital status** | | | | | | | | |
| Single | 13 (8.2) | 73 (5.8) | 1.00 | | | 1.00 | | |
| Married | 131 (82.9) | 1,035 (82.9) | 0.71 | 0.38–1.32 | 0.278 | 0.74 | 0.38–1.45 | 0.375 |
| Ever married | 14 (8.9) | 140 (11.2) | 0.56 | 0.25–1.26 | 0.161 | 0.55 | 0.23–1.32 | 0.181 |
| **Family members** (people) | | | | | | | | |
| ≤ 4 | 81 (51.3) | 641 (51.4) | 1.00 | | | 1.00 | | |
| 5–8 | 64 (40.5) | 522 (41.8) | 0.97 | 0.69–1.37 | 0.865 | 0.90 | 0.62–1.30 | 0.567 |
| ≥9 | 13 (8.2) | 85 (6.8) | 1.21 | 0.65–2.27 | 0.551 | 1.17 | 0.60–2.28 | 0.651 |
| **Tribe** | | | | | | | | |
| Karen | 18 (11.4) | 207 (16.6) | 1.00 | | | 1.00 | | |
| Akha | 51 (32.3) | 357 (28.6) | 1.64 | 0.94–2.89 | 0.084 | 2.03 | 1.03–3.99 | 0.041** |
| Lahu | 19 (12.0) | 217 (17.4) | 1.01 | 0.51–1.97 | 0.984 | 0.97 | 0.46–2.01 | 0.924 |
| Hmong | 24 (15.2) | 174 (13.9) | 1.59 | 0.83–3.02 | 0.160 | 1.49 | 0.74–2.99 | 0.269 |
| Yao | 21 (13.3) | 170 (13.6) | 1.42 | 0.73–2.75 | 0.298 | 1.54 | 0.77–3.10 | 0.224 |
| Lisu | 25 (15.8) | 123 (9.9) | 2.34 | 1.23–4.46 | 0.010* | 2.20 | 1.10–4.42 | 0.027** |
| **Religion** | | | | | | | | |
| Buddhist | 87 (55.1) | 629 (50.4) | 1.21 | 0.87–1.68 | 0.270 | 1.50 | 0.97–2.31 | 0.070 |
| Christian or Muslim | 71 (44.9) | 619 (49.6) | 1.00 | | | 1.00 | | |
| **Education** | | | | | | | | |
| Never attended a school | 104 (65.8) | 871 (69.8) | 0.82 | 0.51–1.33 | 0.421 | 0.74 | 0.40–1.37 | 0.336 |
| Primary school | 31 (17.5) | 219 (19.6) | 0.97 | 0.55–1.73 | 0.924 | 1.01 | 0.53–1.90 | 0.985 |
| Secondary school and higher | 23 (14.6) | 158 (12.7) | 1.00 | | | 1.00 | | |
| **Occupation** | | | | | | | | |
| Unemployed | 29 (18.4) | 161 (12.9) | 1.59 | 0.95–2.64 | 0.075 | 1.62 | 0.94–2.80 | 0.082 |
| Agriculturist | 85 (53.8) | 691(55.4) | 1.06 | 0.72–1.57 | 0.737 | 1.03 | 0.67–1.59 | 0.891 |
| Daily employment or trader | 44 (27.8) | 396 (31.7) | 1.00 | | | 1.00 | | |
| **Annual income** (baht) | | | | | | | | |
| ≤ 50,000 | 108 (68.4) | 841 (67.4) | 0.72 | 0.42–1.24 | 0.234 | 0.72 | 0.41–1.39 | 0.271 |
| 50,001–100,000 | 32 (20.3) | 306 (24.5) | 0.58 | 0.32–1.10 | 0.092 | 0.54 | 0.28–1.04 | 0.066 |
| ≥ 100,001 | 18 (11.4) | 101 (8.1) | 1.00 | | | 1.00 | | |
| **Family debt** | | | | | | | | |
| No | 97 (61.4) | 765 (61.3) | 1.00 | | | 1.00 | | |
| Yes | 61 (38.6) | 483 (38.7) | 0.99 | 0.71–1.40 | 0.982 | 1.16 | 0.79–1.70 | 0.440 |
| **Paternal DM history** | | | | | | | | |
| No | 95 (60.1) | 799 (64.0) | 1.00 | | | 1.00 | | |
| Yes | 7 (4.4) | 68 (5.4) | 0.87 | 0.39–1.94 | 0.726 | 0.78 | 0.33–1.84 | 0.568 |
| Do not know | 56 (35.4) | 381 (30.5) | 1.24 | 0.87–1.76 | 0.238 | 1.62 | 0.83–3.15 | 0.157 |
| **Maternal DM history** | | | | | | | | |

*(Continued)*

**Table 2.** (Continued)

| Factors | Prediabetes mellitus | | Univariate analysis | | | Multivariate analysis | | |
|---|---|---|---|---|---|---|---|---|
| | Yes n (%) | No n (%) | OR | 95% CI | p value | AOR | 95% CI | p value |
| No | 100 (63.3) | 795 (63.7) | 1.00 | | | 1.00 | | |
| Yes | 10 (6.3) | 92 (7.4) | 0.86 | 0.44–1.71 | 0.676 | 0.75 | 0.36–1.55 | 0.432 |
| Do not know | 48 (30.4) | 361 (28.9) | 1.06 | 0.73–1.52 | 0.766 | 0.73 | 0.37–1.46 | 0.375 |
| **Smoking** | | | | | | | | |
| No | 109 (69.0) | 948 (76.0) | 1.00 | | | 1.00 | | |
| Yes | 49 (31.0) | 300 (24.0) | 1.42 | 0.99–2.04 | 0.057 | 1.65 | 0.98–2.61 | 0.053 |
| **Alcohol consumption** | | | | | | | | |
| No | 111 (70.3) | 923 (74.0) | 1.00 | | | 1.00 | | |
| Yes | 47 (29.7) | 325 (26.0) | 1.20 | 0.84–1.73 | 0.320 | 0.92 | 0.57–1.49 | 0.727 |
| **Exercise** | | | | | | | | |
| No | 99 (62.7) | 745 (59.7) | 2.58 | 1.02–6.49 | 0.044* | 2.70 | 1.00–7.12 | 0.051 |
| Sometimes | 54 (34.2) | 406 (32.5) | 2.58 | 1.01–6.23 | 0.049* | 2.56 | 0.95–6.90 | 0.063 |
| Regularly | 5 (3.2) | 97 (7.8) | 1.00 | | | 1.00 | | |
| **BMI** | | | | | | | | |
| Normal weight | 44 (27.8) | 390 (31.2) | 1.00 | | | 1.00 | | |
| Underweight | 9 (5.7) | 60 (4.8) | 1.33 | 0.62–2.86 | 0.467 | 1.41 | 0.62–3.18 | 0.410 |
| Overweight | 105 (66.5) | 798 (63.9) | 1.17 | 0.80–1.69 | 0.418 | 1.23 | 0.83–1.84 | 0.303 |
| **Hypertension** | | | | | | | | |
| No | 100 (63.3) | 895 (71.7) | 1.00 | | | 1.00 | | |
| Yes | 58 (36.7) | 353 (28.3) | 1.47 | 1.04–2.08 | 0.029* | 1.47 | 1.03–2.12 | 0.036** |
| **Total cholesterol** | | | | | | | | |
| Normal | 105 (66.5) | 666 (53.4) | 1.73 | 1.22–2.45 | 0.002* | 2.42 | 1.59–3.67 | <0.001** |
| High | 53 (33.5) | 582 (46.6) | 1.00 | | | 1.00 | | |
| **Triglycerides** | | | | | | | | |
| Normal | 84 (53.2) | 750 (60.1) | 1.00 | | | 1.00 | | |
| High | 74 (46.8) | 498 (39.9) | 1.33 | 0.95–1.85 | 0.095 | 1.79 | 1.21–2.66 | 0.004** |
| **HDL-C** | | | | | | | | |
| Normal | 67 (42.4) | 569 (45.6) | 1.00 | | | 1.00 | | |
| Low | 91 (57.6) | 679 (54.4) | 1.14 | 0.82–1.59 | 0.448 | 0.97 | 0.64–1.46 | 0.867 |
| **LDL-C** | | | | | | | | |
| Normal | 40 (25.3) | 384 (30.8) | 1.00 | | | 1.00 | | |
| High | 118 (74.7) | 864 (69.2) | 1.32 | 0.90–1.91 | 0.160 | 1.90 | 1.24–2.91 | 0.003** |

N/A = Not applicable

* Significance level at α = 0.05

** Significance level at α = 0.05 after controlling for sex and age.

After controlling for age and sex in the multivariate analysis, five factors were found to be associated with prediabetes. Akha and Lisu tribe members had 2.03 (95% CI = 1.03–3.99) and 2.20 (95% CI = 1.10–4.42) times higher odds of having prediabetes than Karen members, respectively. Participants who had HT had 1.47 (95% CI = 1.03–2.08) times higher odds of having prediabetes than those who had normal blood pressure. Those who had a normal total cholesterol level had 2.43 (95% CI = 1.65–3.58) times higher odds of having prediabetes than those who had a high total cholesterol level. Those who had a high triglyceride level had 1.64 (95% CI = 1.16–2.32) times higher odds of having prediabetes than those who had a normal

triglyceride level. Those who had a high LDL-C level had 1.96 (95% CI = 1.30–2.96) times higher odds of having prediabetes than those who had a normal LDL-C level (Table 2).

## Discussion

The hill tribe people between 30 and 59 years old in Thailand have a low socioeconomic status, a low education level, and a low income and work in unskilled jobs. One-fourth of them consumed alcohol and smoked, while only a few people practiced regular exercise. A large proportion were overweight, had an abnormal lipid profile, and suffered from HT. The overall prevalence of prediabetes was 11.2%. Several factors were found to be associated with prediabetes: triglycerides, total cholesterol, LDL-C and HT.

The prevalence of prediabetes among hill tribe people between 30 and 59 years old was 11.2%. The rates of prediabetes among different populations and different countries vary greatly: 22.9% among Bangladeshi people 35 years and older [18], 40.9% among people between 18 and 70 years old in China [19], 35.0% among the adult Omani population [20], 52.9% among Vietnamese people between 45 and 69 years old [21], and 32.2% among Thai adults 35–65 years old [14]. The differences in the prevalence could be due to the differences in the target populations and the methods used for identifying prediabetes. Most studies used fasting blood glucose, but in our study, we used HbA1c to classify prediabetes, which is much more accurate than other methods [2].

Even though the prevalence of prediabetes was lower than that in other populations, several individual characteristics of the participants were associated with a seriously high risk of diabetes, and the next step was the development of prediabetes [1, 3]: 64.2% of the participants were overweight, 92.5% reported nonregular exercise, 24.8% smoked, 26.5% consumed alcohol, 29.2% had HT, and a large proportion had high levels of triglycerides, total cholesterol and LDL-C. These common profiles indicate that hill tribe people in Thailand are at high risk of diabetes and will require large medical expenditures in the future for care and case management [8].

Moreover, in our study, the prevalence did not differ according to sex, age, or tribe. Other studies reported that the prevalence of prediabetes was different between sexes [22], among different age categories [23], and among different tribes [24]. This indicates that the hill tribe people in Thailand share some common characteristics, particularly lifestyle and cooking practices, which include eating behaviors. The relationship between prediabetes and genetic variation among hill tribe people should be investigated.

However, Akha and Lisu people were found to have greater odds of having prediabetes than Karen people. Even though many general lifestyles and behaviors among the hill tribe people are similar, certain factors have some influence on prediabetes in these populations. For instance, alcohol use among Lisu men and women is common, but Karen women and Akha women do not use alcohol [10, 25, 26]. While smoking behavior is commonly found among Akha men and women, Lisu women and Karen women do not smoke [27]. Lisu and Akha cooking practices are similar to Chinese cooking practices, which involve oil, while Karen people use less oil in their daily cooking practice [28, 29]. Certain health-related behaviors and cooking practices can influence the development of prediabetes in these populations.

This study confirms the findings of previous studies that indicated that triglyceride and LDL-C levels were associated with prediabetes. The Canadian Diabetes Association reported that high LDL-C and triglyceride levels were risk factors for prediabetes and diabetes [30]. A study in Vietnam showed that high levels of triglycerides and LDL-C were associated with prediabetes [31]. Moreover, a study in Bangladesh clearly demonstrated that high levels of triglycerides and LDL-C were associated with prediabetes [32]. Some studies [28, 29] conducted

among the hill tribes reported that a large proportion of the hill tribe people had high LDL-C levels and hypertriglyceridemia. Clearly, LDL-C and total cholesterol are related to each other [33], and the association between total cholesterol and prediabetes might be related to the impact of LDL-C. This study showed that a normal total cholesterol level was associated with prediabetes. This could be due to an interruption of LDL-C metabolism. Further investigation found that high total cholesterol levels (normally classified) had a strong correlation with increased LDL-C (r = 0.459, p value≤0.001). Therefore, the association between total cholesterol and prediabetes was a proxy association with LDL-C.

A few limitations were present in this study. First, some participants did not clearly understand Thai, which would impact their ability to answer the questions and understand the NPO instructions. However, we carefully monitored the data, while the village health volunteers helped these participants complete the questionnaire. Due to the nature of a cross-sectional study, some information was very difficult to obtain, such as a history of parental DM, because a large proportion of previous generations did not visit a hospital for health problems. Finally, only 11.2% of participants had prediabetes in the study, which might impact the ability of the statistical model to identify the potential factors associated with prediabetes.

## Conclusion

Hill tribe people between 30 and 59 years old have low socioeconomic status and a high rate of prediabetes, which could progress to diabetes in the future. A large proportion had abnormal lipid profiles and exercised infrequently, which are risk factors for diabetes. More than half of the patients with prediabetes presented with HT, which is another key contributor to diabetes. The need to develop and implement public health interventions that focus on healthy food and dietary behaviors and participation in physical activity, especially regular exercise, to address prediabetes is urgent. Moreover, it is necessary to ensure that health education and essential health messages are delivered effectively to hill tribe people with a low education level. Abnormal lipid parameters relevant to health problems in this population should be further investigated.

## Supporting information

**S1 Appendix. Questionnaire used in the study (English).**
(DOCX)

**S2 Appendix. Data file for the study.**
(XLSX)

## Acknowledgments

The article processing charge of this work was financially supported by Mae Fah Luang University. We would like to thank all the village headmen for supporting this work in their villages. We would also like to thank all the participants for providing essential information for the study.

## Author Contributions

**Conceptualization:** Tawatchai Apidechkul.

**Data curation:** Tawatchai Apidechkul, Chalitar Chomchiei, Panupong Upala, Ratipark Tamornpark.

**Formal analysis:** Tawatchai Apidechkul, Chalitar Chomchiei, Panupong Upala, Ratipark Tamornpark.

**Funding acquisition:** Tawatchai Apidechkul.

**Investigation:** Tawatchai Apidechkul, Chalitar Chomchiei, Panupong Upala, Ratipark Tamornpark.

**Methodology:** Tawatchai Apidechkul.

**Project administration:** Tawatchai Apidechkul.

**Writing – original draft:** Tawatchai Apidechkul, Chalitar Chomchiei, Panupong Upala, Ratipark Tamornpark.

**Writing – review & editing:** Tawatchai Apidechkul, Chalitar Chomchiei, Panupong Upala, Ratipark Tamornpark.

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
