## [Decision Letter · Decision Letter 0]

16 Aug 2021

PONE-D-21-21401

Epidemiology of prediabetes mellitus among hill tribe adults in Thailand

PLOS ONE

Dear Dr. Apidechkul,

Thank you for submitting your manuscript to PLOS ONE. After careful consideration, we feel that it has merit but does not fully meet PLOS ONE’s publication criteria as it currently stands. Therefore, we invite you to submit a revised version of the manuscript that addresses the points raised during the review process.

I have received the reports from our advisors on your manuscript which you submitted to PLOS ONE.

Based on the comments received, I feel that your manuscript could be reconsidered for publication should you be prepared to incorporate major revisions.

When preparing your revised manuscript, you are asked to carefully consider the reviewer comments below and submit a list of responses to the comments.

Editor Comments: The paper should be checked by a professional speaker of English before complete acceptance.

We look forward to receiving your revised manuscript.

Kind regards,

Muhammad Sajid Hamid Akash

Academic Editor

PLOS ONE

Journal Requirements:

Furthermore please provide additional information regarding how participants were recruited for the study, the recruitment date range (month and year) and a descriptions of where participants were recruited and where the research took place.

Reviewers' comments:

Reviewer's Responses to Questions

**Comments to the Author**

1. Is the manuscript technically sound, and do the data support the conclusions?

Reviewer #1: Yes

2. Has the statistical analysis been performed appropriately and rigorously? 

Reviewer #1: Yes

3. Have the authors made all data underlying the findings in their manuscript fully available?

Reviewer #1: Yes

4. Is the manuscript presented in an intelligible fashion and written in standard English?

Reviewer #1: Yes

5. Review Comments to the Author

Reviewer #1: add the duration time to study. add the detail of sampling technique of study, how to select study subjects. In statistical analysis, add the detail of method of multivariable logistic regression. Table2, select OR=1 at the first line of every variable, add the method of multivariable technique at the end of Table2. Check the format style and correction of the references.

6. PLOS authors have the option to publish the peer review history of their article (what does this mean?). If published, this will include your full peer review and any attached files.

Reviewer #1: No

---

## [Author Response · Author response to Decision Letter 0]

24 Sep 2021

Response to editor and reviewers’ comments

Dear Editor,

Thank you very much for the opportunity to submit the revised version. We have revised all points as editor comments in this version. The manuscript has been double checked by the American Journal Experts (AJE) with the reference No. No. 85D4-4E23-480D-5FA2-42C9.

Thank you,

TK

PONE-D-21-21401

Epidemiology of prediabetes mellitus among hill tribe adults in Thailand

PLOS ONE

Dear Dr. Apidechkul,

Thank you for submitting your manuscript to PLOS ONE. After careful consideration, we feel that it has merit but does not fully meet PLOS ONE’s publication criteria as it currently stands. Therefore, we invite you to submit a revised version of the manuscript that addresses the points raised during the review process.

I have received the reports from our advisors on your manuscript which you submitted to PLOS ONE.

Based on the comments received, I feel that your manuscript could be reconsidered for publication should you be prepared to incorporate major revisions.

When preparing your revised manuscript, you are asked to carefully consider the reviewer comments below and submit a list of responses to the comments.

Editor Comments: The paper should be checked by a professional speaker of English before complete acceptance.

: This manuscript has been double checked by the American Journal Experts (AJE) with No. 85D4-4E23-480D-5FA2-42C9. 

We look forward to receiving your revised manuscript.

Kind regards,

Muhammad Sajid Hamid Akash

Academic Editor

PLOS ONE

Journal Requirements:

: Thank you, we have checked. 

Furthermore please provide additional information regarding how participants were recruited for the study, the recruitment date range (month and year) and a descriptions of where participants were recruited and where the research took place.

: Questionnaire used has been uploaded.

: It’s moved in proper place.

Reviewers' comments:

Reviewer's Responses to Questions

Comments to the Author

1. Is the manuscript technically sound, and do the data support the conclusions?

Reviewer #1: Yes

2. Has the statistical analysis been performed appropriately and rigorously? 

Reviewer #1: Yes

3. Have the authors made all data underlying the findings in their manuscript fully available?

Reviewer #1: Yes

4. Is the manuscript presented in an intelligible fashion and written in standard English?

Reviewer #1: Yes

5. Review Comments to the Author

Reviewer #1: add the duration time to study. add the detail of sampling technique of study, how to select study subjects. In statistical analysis, add the detail of method of multivariable logistic regression. Table2, select OR=1 at the first line of every variable, add the method of multivariable technique at the end of Table2. Check the format style and correction of the references.

6. PLOS authors have the option to publish the peer review history of their article (what does this mean?). If published, this will include your full peer review and any attached files.

Do you want your identity to be public for this peer review? For information about this choice, including consent withdrawal, please see our Privacy Policy.

Reviewer #1: No

Thank you,

TK

Assistant Professor Dr.Tawatchai Apidechkul

Deputy Dean, School of Health Science, MFU

Director, Center of Excellence of the Hill tribe Health Research, WHO-CC 

Former Hubert H Humphrey Fellow (2013-2014), Emory University

Global Health Delivery Intensive (Harvard School of Public Health)

---

## [Decision Letter · Decision Letter 1]

19 Apr 2022

PONE-D-21-21401R1Epidemiology of prediabetes mellitus among hill tribe adults in ThailandPLOS ONE

Dear Dr. Apidechkul,

Thank you for submitting your manuscript to PLOS ONE. After careful consideration, we feel that it has merit but does not fully meet PLOS ONE’s publication criteria as it currently stands. Therefore, we invite you to submit a revised version of the manuscript that addresses the points raised during the review process. Specifically, the issues with methodology raised by the reviewers. 

We look forward to receiving your revised manuscript.

Kind regards,

Xi Pan

Academic Editor

PLOS ONE

Reviewers' comments:

Reviewer's Responses to Questions

**Comments to the Author**

1. If the authors have adequately addressed your comments raised in a previous round of review and you feel that this manuscript is now acceptable for publication, you may indicate that here to bypass the “Comments to the Author” section, enter your conflict of interest statement in the “Confidential to Editor” section, and submit your "Accept" recommendation.

Reviewer #2: (No Response)

Reviewer #3: All comments have been addressed

2. Is the manuscript technically sound, and do the data support the conclusions?

Reviewer #2: Yes

Reviewer #3: Yes

3. Has the statistical analysis been performed appropriately and rigorously? 

Reviewer #2: No

Reviewer #3: Yes

4. Have the authors made all data underlying the findings in their manuscript fully available?

Reviewer #2: No

Reviewer #3: Yes

5. Is the manuscript presented in an intelligible fashion and written in standard English?

Reviewer #2: Yes

Reviewer #3: Yes

6. Review Comments to the Author

Reviewer #2: Thank you for inviting me for this review. This is an interesting community-based cross-sectional study, and the first hand in-field data collection was valuable and provided some insights into the prevention and epidemiological understanding of prediabetes mellitus in Thailand. I hope my comments below are useful considerations.

1. Could you please clarify when the survey was conducted and time span of this study?

2. Statistical analysis section:

• Correcting “SPS program” with “SPSS program”? A typo maybe.

• Please clarify/specify your multivariable logistic model. The author mentioned that age and gender were the only two controlled confounders. Based on the results, it seems other variables were also included/controlled in the final multivariable model, while only the significant four variables were reported. Please provide a complete list of variables that you had used for this model. Based on the authors’ description of previous studies, it seems BMI, alcohol consumption and other socioeconomic factors all have impacts on the outcome to certain level. Therefore, please provide the rationale about your variable selection, in terms of model performance, data quality and clinical meanings.

• Any sampling weights applied in your data analysis since you have surveyed different geographical locations (different hill tribes)?

• In Table 2, could you please present the proportion of each factor for the patients with prediabetes mellitus and patients without prediabetes mellitus? Then we have a clear picture about how the demographical/socioeconomic and medical factors were distributed in each cohort, for instance, to see if the gender decomposition is similar in the patients with prediabetes mellitus versus the patients without prediabetes mellitus.

• It seems the two cohorts (the patients with prediabetes mellitus versus the patients without prediabetes mellitus) were not propensity score matched/ approximately matched cohorts. Please include this as a limitation/address the possible impacts for your model.

3. Please check the English writing grammar.

4. I noticed the author published a similar paper in 2018:

"Apidechkul T. Prevalence and factors associated with type 2 diabetes mellitus and hypertension among the hill tribe elderly populations in northern Thailand. BMC Public Health. 2018 Jun 5;18(1):694. doi: 10.1186/s12889-018-5607-2. PMID: 29871598; PMCID: PMC5989444."

Could please also provide participants’ selection flowchart in current study to see how the weighting sampling method could be applied if that is possible? The reason to do this is that I am wondering if any geographical impacts applied/pre-exists for the prevalence of prediabetes mellitus. After reading this similar work, I gained some background information in your study location and it seems the different hill tribe had various prevalence of type 2 diabetes mellitus and hypertension, it made me pondering if that case was also applied for prevalence of prediabetes mellitus.

Reviewer #3: This interesting study investigated the prevalence of prediabetes among subjects from six hill tribes in Thailand, which answered an important research question. The manuscript is generally well written and clearly presented. I only have a few comments for the authors to consider.

1. This study found that people having a normal total cholesterol level are more likely to have prediabetes than those having a high cholesterol level. This is controversial in the literature. Although the authors provided some discussion around this finding, I would like to recommend the authors add more discussion around the potential confounding and the relation between LDL-C, HDL-C, and total cholesterol. Please also consider adding a few references here.

2. I am wondering why only age and sex were adjusted as covariates in the multivariable analysis. Since there are six tribes included in this study and the authors mentioned each of the tribes has its own culture, it seems tribe is one of the confounders. Please consider adding some explanations or discussions.

3. In table 2, please consider providing all the odds ratios from multivariable analyses, even if some are not statistically significant.

7. PLOS authors have the option to publish the peer review history of their article (what does this mean?). If published, this will include your full peer review and any attached files.

Reviewer #2: No

Reviewer #3: **Yes: **Junjie Ma

---

## [Author Response · Author response to Decision Letter 1]

9 Jun 2022

Response to reviewers’ comments

Review Comments to the Author

Reviewer #2: Thank you for inviting me for this review. This is an interesting community-based cross-sectional study, and the first hand in-field data collection was valuable and provided some insights into the prevention and epidemiological understanding of prediabetes mellitus in Thailand. I hope my comments below are useful considerations.

1. Could you please clarify when the survey was conducted and time span of this study?

: Data were collected between November 2019 and March 2020; please see the abstract and methods sections (page 6, lines 16-17).

2. Statistical analysis section:

• Correcting “SPS program” with “SPSS program”? A typo maybe.

: Thank you, it has been corrected.

• Please clarify/specify your multivariable logistic model. The author mentioned that age and gender were the only two controlled confounders. Based on the results, it seems other variables were also included/controlled in the final multivariable model, while only the significant four variables were reported. Please provide a complete list of variables that you had used for this model. Based on the authors’ description of previous studies, it seems BMI, alcohol consumption and other socioeconomic factors all have impacts on the outcome to certain level. Therefore, please provide the rationale about your variable selection, in terms of model performance, data quality and clinical meanings.

: We started with the univariate analysis by having one independent variable and dependent variable and set the significance threshold at �=0.05, and any variable with a p value equal to or less than 0.05 was considered significant. This step of the analysis was repeated until all independent variables were completed. Afterward, all independent variables were added to the model with the dependent variable, and the significance was assessed. The least significant variable (with the greatest p value) was removed from the model, and the Hosmer Lemeshow Chi-square test was used to assess the goodness of fit (nonsignificant). The process of testing the model was repeated by removing all nonsignificant variables in the model and showing that the Hosmer Lemeshow Chi-square test result was nonsignificant, which was the final model. However, before interpretation, age and sex were adjusted in the model to control their effects as confounding factors.

Of course, with the conditions (both exposures (independent variables) and outcomes (disease) examined at the same time) of the cross-sectional study used in this project, which is intended to assess the prevalence and predict the factors associated with the outcome, the association detected might not be fully accurate, similar to other stronger study designs that focus only on testing the association.

During the analysis, we examined all independent variables with the outcome (pre-DM) and retained some variables as the best predictors in the model before making the interpretation.

We have added the essential information to the statistical analysis section, page 6 line 32; and page 7 lines 1-3. 

• Any sampling weights applied in your data analysis since you have surveyed different geographical locations (different hill tribes)?

: Thank you for the great comment. We used five villages from each tribe to select the participants for the study. All people who met the criteria and lived in one of the five selected villages for each tribe were invited to participate in the study. Finally, with the number of people living in each village, the proportion of participants from each tribe was still reflected in the study population:

Karen 16%, Hmong 14%, Yao 13%, Akha 29%, Lahu 16.8%, and Lisu 10.5%.

Even the actual population in the six different tribes are a bit different: in 2019, there were 749 hill tribe villages in Chiang Rai Province, Thailand, which included 316 Lahu villages (51,339 persons (26.5%)), 243 Akha villages (74,403 persons (38.5%)), 63 Yao villages (16,227 persons (8.4%)), 56 Hmong villages (33,478 persons (17.0%)), 36 Karen villages (7,933 persons (4%)), and 35 Lisu villages (9,632 persons (4.9%)) [12]. Page 3, lines 20-23.

: We have carefully considered this excellent point and found that the statistics shown in Table 2, particularly the CIs, had high power, which means that the sample size was large enough for testing the hypothesis and that the proportions could well reflect the different sizes of the populations of each tribe. However, during sample size calculation, we did not consider this idea, and it will be used in our next project to ensure that the final model can accurately reflect the hill tribe population. Thank you so much.

• In Table 2, could you please present the proportion of each factor for the patients with prediabetes mellitus and patients without prediabetes mellitus? Then we have a clear picture about how the demographical/socioeconomic and medical factors were distributed in each cohort, for instance, to see if the gender decomposition is similar in the patients with prediabetes mellitus versus the patients without prediabetes mellitus.

: Yes, please see columns 2 and 3, which have been changed to the % column. Then, we can see the proportion distribution in each factor between those who had pre-DM and those who did not.

• It seems the two cohorts (the patients with prediabetes mellitus versus the patients without prediabetes mellitus) were not propensity score matched/ approximately matched cohorts. Please include this as a limitation/address the possible impacts for your model.

: Thank you. We completely agree with you and have included this as one of the key limitations of the study.

3. Please check the English writing grammar.

: Thank you. The English has been checked by American Journal Experts (AJE) with reference no. 85D4-4E23-480D-5FA2-42C9 . 

4. I noticed the author published a similar paper in 2018:

"Apidechkul T. Prevalence and factors associated with type 2 diabetes mellitus and hypertension among the hill tribe elderly populations in northern Thailand. BMC Public Health. 2018 Jun 5;18(1):694. doi: 10.1186/s12889-018-5607-2. PMID: 29871598; PMCID: PMC5989444."

: The two projects are different. The first project you mentioned, which was published previously in BMC Public Health, was designed to identify factors that contributed to DM and HT in the elderly population. In current project, however, we focused on people between 30 and 59 years old. The current project was performed after our first project was completed, and we extended our ideas in the second project. The first project was supported by the National Research Council of Thailand, while the second project was supported by the Health System Research Institute, Thailand (Grant No 61-027). Therefore, the two papers are different.

Could please also provide participants’ selection flowchart in current study to see how the weighting sampling method could be applied if that is possible? The reason to do this is that I am wondering if any geographical impacts applied/pre-exists for the prevalence of prediabetes mellitus. After reading this similar work, I gained some background information in your study location and it seems the different hill tribe had various prevalence of type 2 diabetes mellitus and hypertension, it made me pondering if that case was also applied for prevalence of prediabetes mellitus.

: Thank you for the comment. Please see Figure 1 on page 4.

Reviewer #3: This interesting study investigated the prevalence of prediabetes among subjects from six hill tribes in Thailand, which answered an important research question. The manuscript is generally well written and clearly presented. I only have a few comments for the authors to consider.

1. This study found that people having a normal total cholesterol level are more likely to have prediabetes than those having a high cholesterol level. This is controversial in the literature. Although the authors provided some discussion around this finding, I would like to recommend the authors add more discussion around the potential confounding and the relation between LDL-C, HDL-C, and total cholesterol. Please also consider adding a few references here.

: Thank you for the great comment. We have added information including references in this section; please see page 16, lines 12-16.

2. I am wondering why only age and sex were adjusted as covariates in the multivariable analysis. Since there are six tribes included in this study and the authors mentioned each of the tribes has its own culture, it seems tribe is one of the confounders. Please consider adding some explanations or discussions.

: Thank you for the great comment. From our literature review, we found that age and sex were major confounding factors in predicting diabetes and prediabetes.

: We also reanalyzed entire steps and used the Karen tribe as the reference (because Karen had the lowest prevalence) and found that two tribes, namely, Akha and Lisu, had significantly greater odds of developing pre-DM. Thank you so much for noting this. We have added this information to the discussion section as well.

3. In table 2, please consider providing all the odds ratios from multivariable analyses, even if some are not statistically significant.

: We have added all ORs in the multivariate model; please see table 2.

Thank you so much.

TK

Assist Prof. Dr. Tawatchai Apidechkul, MSc (Infectious Epidemiology), Dr. P. H (Epidemiology)

Dean, School of Health Science, Mae Fah Luang University

Director, Center of Excellence of Hill Tribe Health Research, WHO-CC

Former Hubert H Humphrey Fellow (2013-2014), Emory University

Global Health Delivery Intensive (Harvard School of Public Health)

---

## [Decision Letter · Decision Letter 2]

11 Jul 2022

Epidemiology of prediabetes mellitus among hill tribe adults in Thailand

PONE-D-21-21401R2

Dear Dr. Apidechkul,

We’re pleased to inform you that your manuscript has been judged scientifically suitable for publication and will be formally accepted for publication once it meets all outstanding technical requirements.

Kind regards,

Xi Pan

Academic Editor

PLOS ONE

Additional Editor Comments (optional):

Reviewers' comments:

Reviewer's Responses to Questions

**Comments to the Author**

1. If the authors have adequately addressed your comments raised in a previous round of review and you feel that this manuscript is now acceptable for publication, you may indicate that here to bypass the “Comments to the Author” section, enter your conflict of interest statement in the “Confidential to Editor” section, and submit your "Accept" recommendation.

Reviewer #2: All comments have been addressed

Reviewer #3: All comments have been addressed

2. Is the manuscript technically sound, and do the data support the conclusions?

Reviewer #2: Yes

Reviewer #3: Yes

3. Has the statistical analysis been performed appropriately and rigorously? 

Reviewer #2: Yes

Reviewer #3: Yes

4. Have the authors made all data underlying the findings in their manuscript fully available?

Reviewer #2: Yes

Reviewer #3: Yes

5. Is the manuscript presented in an intelligible fashion and written in standard English?

Reviewer #2: Yes

Reviewer #3: Yes

6. Review Comments to the Author

Reviewer #2: I would like to thank the authors for addressing my initial comments. The authors have provided a nicely detailed and thorough response to the comments from the previous review and have addressed my major concerns regarding the survey administration and statistical analysis plan. Following the revision to the article, the paper had been sufficiently improved.

Reviewer #3: All my comments have been addressed in this version. This interesting study will contribute the research area of diabetes.

7. PLOS authors have the option to publish the peer review history of their article (what does this mean?). If published, this will include your full peer review and any attached files.

Reviewer #2: No

Reviewer #3: **Yes: **Junjie Ma

---

## [Editor Report · Acceptance letter]

14 Jul 2022

PONE-D-21-21401R2 

Epidemiology of prediabetes mellitus among hill tribe adults in Thailand 

Dear Dr. Apidechkul:

I'm pleased to inform you that your manuscript has been deemed suitable for publication in PLOS ONE. Congratulations! Your manuscript is now with our production department. 

Kind regards, 

on behalf of

Dr. Xi Pan 

Academic Editor

PLOS ONE